# Clinical impact of respiratory virus in pulmonary exacerbations of children with Cystic Fibrosis

**Viviane Mauro Correa Meyer**[1¤*], **Marilda Mendonça Siqueira**[1], **Patricia Fernandes Barreto Machado Costa**[2,3], **Braulia Costa Caetano**[1], **Jonathan Christian Oliveira Lopes**[1], **Tânia Wrobel Folescu**[2], **Fernando do Couto Motta**[1]

1 Respiratory Virus and Measles Laboratory, Oswaldo Cruz Institute - IOC/FIOCRUZ, Rio de Janeiro, Brazil, 2 Pulmonology Department, National Institute of Women, Children and Adolescents Health Fernandes Figueira - IFF/FIOCRUZ, Rio de Janeiro, Brazil, 3 Pediatrics Department, Federal University of the State of Rio de Janeiro - UNIRIO, Rio de Janeiro, Brazil

¤ Current address: Pediatric Emergency Department Hospital Israelita Albert Einstein, São Paulo, Brazil
* vivianecorreameyer@gmail.com

## Abstract

### Backgrounds

Cystic Fibrosis (CF) is a genetic, multisystemic, progressive illness that causes chronic suppurative lung disease. A major cause of morbimortality in this condition are pulmonary exacerbations. Although classically attributed to bacterial infections, respiratory virus have been increasingly recognized in its ethiopathogeny.

### Methods

Nasopharyngeal swab samples were collected from children < 18 years old with CF in Rio de Janeiro, Brazil, with pulmonary exacerbation criteria. Samples were submitted to RT-PCR for Adenovirus, Influenza A and B, Parainfluenza Virus, Respiratory Syncytial Virus (RSV), Metapneumovirus and Rhinovirus. Virus positive and virus negative groups were compared in regards to clinical presentation, severity of exacerbation and bacterial colonization.

### Results

Out of 70 samples collected from 48 patients, 35.7% were positive for respiratory viruses. Rhinovirus were the most common (28% of all positive samples), followed by RSV. The virus positive group was associated with change in sinus discharge (p = 0.03). Considering only patients younger than five years old, positive virus detection was also associated with fever (p = 0.01). There was no significant difference in clinical severity or in bacterial colonization between virus positive and negative groups.

**Data Availability Statement:** All relevant data are within the manuscript and its Supporting Information files.

**Funding:** This study was supported by the Brazilian Ministry of Health and Fundação Oswaldo Cruz - FIOCRUZ (projects 25030.000964/2014 and 25030.000932/2017-82; National Health Foundation). The funders had no role in study design, data collection and analysis, decision to publish, or preparation of the manuscript.

**Competing interests:** The authors have declared that no competing interests exist.

## Conclusions

Prospective studies are still needed to assess the long term impact of viral infections in patients with CF, and their interaction with the bacterial microbiome in these patients.

## Introduction

Cystic Fibrosis (CF) is a genetic multisystemic progressive condition, caused by mutations of the Cystic Fibrosis Transmembrane Regulation (CFTR) gene. Pulmonary disease is the main cause of mortality in CF patients [1]. The progressive lung damage is marked by episodes of acute worsening of symptoms, called pulmonary exacerbations (PE), which are associated with disease progression and have an important impact on patients' quality of life [1, 2]. Bacterial infections have been historically acknowledged as the major cause of PE. However, with the popularization of molecular techniques for virus detection, several studies have demonstrated a large prevalence of virus in PE events [3–15], showing significant association between virus detection and PE symptoms [5, 8, 9, 12, 15]. Despite this growing identification, little is still known about the clinical impact of respiratory virus infection in CF patients, with studies often presenting conflicting results.

The aim of this study was to determine the prevalence of respiratory virus infections in children and adolescents with CF during PEs, and compare virus positive and negative groups in regards to clinical manifestations, severity of PE and bacterial colonization.

## Material and methods

CF patients were recruited from the CF service, in Fernandes Figueira National Institute for Women, Children and Adolescent Health at Rio de Janeiro/ Brazil, from January to December 2018. The inclusion criteria were age < 18 years; regular follow up at the CF center and presence of PE, based on the Fuchs Criteria [16]. Shortly, this criteria defines an exacerbation by the presence of at least four of the following signs and symptoms: change in sputum pattern; new or worsening hemoptysis; increase in cough; dyspnea; malaise or fatigue; fever; anorexia or weight loss; sinus pain; change in sinus discharge; change in physical examination of the chest; decrease in pulmonary function by 10% of the forced expiratory volume, and radiographic changes indicative of pulmonary infection. Patients were excluded if they had other chronic pulmonary, cardiovascular, neurologic, digestive, or rheumatologic disease not related to CF.

Clinical data was collected from patients' charts. The Cystic Fibrosis Clinical Score (CFCS) [17] was applied to evaluate PE severity and the Shwachman-Kulczycki Score [18] to evaluate CF severity. When a chest X-ray was not performed at the moment, the last X-ray performed was used to apply this score. Nutritional evaluation was performed according to the Cystic Fibrosis Nutritional Guidelines [19].

Total nucleic acid was extracted from combined nasopharyngeal swabs using QIAamp® Viral RNA mini kit (QIAGEN, Hamburg, Germany), generating 80uL of purified nucleic acid in the final step. The Real Time Reverse Transcription Polymerase Chain Reaction (RT-PCR) for Rhinovirus (RV), Adenovirus (AdV), Respiratory Syncytial Virus (RSV), Influenza A and B, human metapneumovirus (hMpV) and Parainfluenza (PIV) 1, 2 and 3 detection was performed using Go Taq® Probe 1-Step RT-qPCR System kit (Promega, USA) in a ABI 7500 real time thermocycler. The cycling protocol was: 45˚C/30min, 95˚C/5min followed of 45 cycles of

**Table 1. Results of real time RT-PCR respiratory viruses detection.**

| Virus | Positive Samples (n = 25) |
|---|---|
| Rhinovirus | 7 (28%) |
| Respiratory Syncytial Virus | 6 (24%) |
| Adenovirus | 5 (20%) |
| human Metapneumovirus | 4 (16%) |
| Parainfluenza virus | 3 (12%) 1 PIV1, 1 PIV 2, 1 PIV3 |
| Influenza A | 2 (8%) 1 H1N1 and 1 H3N2 |

95˚C/15s, 55˚C/30, with data collection in the last stage. All molecular biology procedures and analysis were performed in the Respiratory Virus and Measles Laboratory at IOC/FIOCRUZ. Real time results were made available to the CF Center's physicians and to the patients right after their analysis.

Oropharyngeal swabs or sputum samples for bacterial culture were also collected at each visit following the CF Center's protocol, and its results were made available to the researches through patients' charts. This study was approved by both institutions' Research Ethics Committee (protocol n. 79277117.0.0000.5269) and informed consent was obtained from all participants and their parents/guardians.

Statistical analysis was performed using the program R, version 3.5.2 ® (2018). Virus positive and negative groups were compared using chi-squared and Fisher's Exact Test for categorical variables, Students T test for normally-distributed numerical variables and Mann-Whitney's U Test for skewed data. Results were considered statistically significant when $p < 0.05$.

## Results

During the year of 2018, 183 patients were followed by the CF Center, totaling 706 appointments. Forty-eight of these patients presented PE during appointments, in 71 different occasions. One case was excluded because the patient had chronic hypoxic ischemic encephalopathy, leaving a total of 47 patients and 70 episodes of PE included in the study. All 70 swab samples were tested by real time RT-PCR, and 25 (35.7%) were positive, as shown in Table 1. Two samples tested positive for more than one virus: one case of PIV 1 + RSV and one of PIV 2 + HMpV. Epidemiological characteristics of the patients included in the study are described in Table 2. Clinical characteristics of virus positive and negative groups are shown in Table 3.

**Table 2. Baseline characteristics of the 47 patients enrolled in the study.**

| Characteristics | Patients (n = 47) |
|---|---|
| Gender (male:female) | 24:23 |
| Homozygus F508del | 12 (25.5%) |
| Heterozygus F508del | 13 (27.6%) |
| Pancreatic Insufficiency | 41 (87.7%) |
| Diabetes Mellitus | 1 (2.1%) |
| Influenza Immunization | 25 (53.2%) |

**Table 3. Clinical and epidemiological characteristics of virus positive and negative groups.**

|  | Virus Positive (n = 25) | Virus Negative (n = 45) | p value |
|---|---|---|---|
| Median age in years (range +- SD) | 3.9 (0.3–17,9; +- 4.9) | 5,5 (0.3–17; +- 4.6) | 0.3[a] |
| Median Days of symptoms (range+- SD) | 6 (1–21; +-5.3) | 6 (0–30; +-5.5) | 0.7 [a] |
| **Clinical Features** |  |  |  |
| Change in Sinus Discharge | 21 (84%) | 26 (57.8%) | 0.03 |
| Worsening cough | 25 (100%) | 44 (97.8%) | 1 |
| Change in sputum pattern | 22 (88%) | 38 (84.4%) | 1 |
| Worsening dyspnea | 14 (56%) | 27 (60%) | 0.8 |
| Hemoptysis | 1 (4%) | 1 (2.2%) | 1 |
| Fatigue | 9 (36%) | 10 (22.2%) | 0.21 |
| Fever | 16 (64%) | 19 (42.2%) | 0.08 |
| Anorexia/weight loss | 14 (56%) | 26 (57.8%) | 0.89 |
| Sinus pain | 1 (4%) | 2 (4.4%) | 1 |
| Low oxygen saturation | 7 (28%) | 15 (33.3%) | 0.65 |
| Abnormal Chest Auscultation | 20 (80%) | 42 (93.3%) | 0.1 |
| **Treatment** |  |  |  |
| Antibiotics | 19 (76%) | 40 (88.9%) | 0.18 |
| Oseltamivir | 1 (4%) | 0 | 0.35 |
| Oxygen therapy | 6 (24%) | 15 (33.3%) | 0.41 |
| Noninvasive Ventilation | 3 (12%) | 4 (8.9%) | 0.69 |
| Hospital Admission | 10 (40%) | 21 (46.7%) | 0.11 |
| Mean length of hospital stay in days (range +- SD) | 14.6 (7–21; +- 3.8) | 16,5 (8–34; +- 5.6) | 0.45 [a] |
| **Severity Scores** |  |  |  |
| Mean CFCS[c] (range +- SD) | 30.3 (19–39; +- 5.4) | 29.4 (18–41; +- 5.9) | 0.25[b] |
| Mean Shwachman-Kulczycki Score (range +- SD) | 72.2 (45–95; +-11.9) | 71.6 (35–90; +-12.4) | 0.42 [b] |

[a]Student's T test.
[b]Mann Whitney's U test.
[c]*Cystic Fibrosis Clinical Score*.

The number of X-ray exams ordered and the finding of new radiologic images suggesting infection did not differ between virus positive and negative groups. Bacterial colonization status previous to the PE events are described in Table 4. The bacterial culture from sputum or oropharyngeal swabs of the moment of the PE were tested for all samples enrolled in the study, but two (Table 5).

**Table 4. Previous bacterial colonization.**

| Previous Bacterial Colonization | Virus Positive (n = 25) | Virus Negative (n = 45) | P value |
|---|---|---|---|
| Negative or *Staphyloccocus aureus* | 6 (24%) | 11 (24.4%) | 0.97 |
| Intermittent MRSA[a] | 3 (12%) | 9 (20%) | 0.52 |
| Intermittent *Pseudomonas aeruginosa* | 12 (48%) | 20 (44.4%) | 0.77 |
| Chronic *Pseudomonas aeruginosa* | 4 (16%) | 4 (8.9%) | 0.44 |
| Intermittent *Burkholderia cepacia* | 3 (12%) | 5 (11.1%) | 1 |
| Chronic *Burkholderia cepacia* | 1 (4%) | 5 (11.1%) | 0.41 |

[a]Methicillin-resistant *Staphylococcus aureus*.

**Table 5. Bacterial culture results from sputum or oropharyngeal swab of virus positive and negative patients.**

| Bacterial Culture Results | Virus Positive (n = 24) | Virus Negative (n = 44) | p Value |
|---|---|---|---|
| Negative | 3 (12.5%) | 7 (15.9%) | 1 |
| S. aureus | 13 (54.2%) | 25 (56.8%) | 0.83 |
| MRSA[a] | 3 (12.5%) | 6 (13.6%) | 1 |
| K. pneumoniae | 0 | 1 (2.3%) | 1 |
| P. aeruginosa | 11 (45.8%) | 12 (27.3%) | 0.12 |
| B. cepacia | 1 (4.2%) | 8 (18.2%) | 0.14 |
| A. xylosoxidans | 1 (4.2%) | 0 | 0.35 |
| S. maltophilia | 2 (8.3%) | 0 | 0.12 |
| K. pneumoniae | 0 | 1 (2.3%) | 1 |
| NFGNB[b] | 1 (4.2%) | 2 (4.5%) | 1 |

[a]Methicillin-resistant *Staphylococcus aureus*.

[b]Non-fermentative Gram-Negative Bacilli, *Pseudomonas* and *Burkholderia* excluded.

**Table 6. Clinical and epidemiological characteristics of virus positive and negative patients < 5 years.**

| Variable | Virus Positive (n = 14) | Virus Negative (n = 22) | p value |
|---|---|---|---|
| Median Age in Years (range +- SD) | 2.3 (0.3–4.9; +-1.3) | 2.6 (0.3–4.3; +-1.2) | 0.29[a] |
| **Clinical and Radiological Features** | | | |
| Change in sinus discharge | 12 (85.7%) | 13 (59.1%) | 0.14 |
| Worsening cough | 14 (100%) | 22 (100%) | 1 |
| Change in sputum pattern | 11 (78.6%) | 15 (68.2%) | 0.71 |
| Worsening dyspnea | 9 (64.3%) | 16 (72.7%) | 0.72 |
| Hemoptysis | 0 | 0 | 1 |
| Fatigue | 6 (42.9%) | 3 (13.6%) | 0,11 |
| Fever | 9 (64.3%) | 5 (22.7%) | 0,01 |
| Anorexia / weight loss | 8 (57.1%) | 11 (50%) | 0,68 |
| Oxygen therapy | 4 (8.6%) | 7 (31.8%) | 1 |
| Abnormal Chest Auscultation | 13 (92.9%) | 21 (95.4%) | 1 |
| Hospital Admission | 6 (42.9%) | 10 (45.4%) | 1 |
| **Severity Scores** | | | |
| CFCS[b] (mean) | 30.3 (19–38; +-5.7) | 28.1 (19–37; +-4.4) | 0.12[a] |
| Mean Shwachman-Kulczycki Score (range +- SD) | 78.2 (65–95; +-8.5) | 76.4 (60–90; +-7.4) | 0.25 [a] |

[a]Student's T test.

[b]Cystic Fibrosis Clinical Score.

A subgroup analysis was conducted with 37 samples from children <5 years. Fourteen samples (37.8%) were positive for virus detection, namely: six RSV, four RV, two hMpV, one Influenza A(H1N1)pdm09, one PIV 1, one AdV. Clinical and epidemiological characteristics of these patients are shown in Table 6.

## Discussion

The prevalence of respiratory virus in this study was 35.7%. In agreement with previous studies, *picornaviridae* viruses were the most frequently detected viruses, regardless of age [4–9, 12, 20, 21]. RSV was the second most frequent virus, and had the lowest hospital admission rate, even though it has been suggested that CF patients could be at risk for more severe infection

by this agent [22]. One possible explanation is that severe infections are more common in patients under two years of age, whereas the median age of the patients in our study was 4.6 years old. Noteworthy, the mean age of RSV positive patients was two years old.

Regarding clinical findings, change in sinus discharge was associated with viral detection (p 0.03), as would be expected and has been described by other authors [8, 9]. In patients under five years old, fever had a significant association with virus detection (p = 0.01), as also previously described [3]. Influenza has been found to be particularly associated with presence of fever [9]. In our study, the two influenza positive cases were older than five years and had this symptom indeed.

Regarding treatment decisions, antibiotic prescription was not significantly different in patients with and without virus detection, corroborating that these situations may be difficult to clinically distinguish. On the other hand, oseltamivir prescription was extremely low, even when clinical symptoms were indicative for its empiric use. Only one patient received the anti-viral treatment, which was started empirically and was later confirmed to have Influenza A H3N2. The second patient with influenza A was not treated because the drug was not started empirically and viral detection results were only available after 48 hours of symptoms, when the patient was already clinically improving. It is important to note that the vaccination rate for influenza was also low (58.1%), as previously reported [14, 15]. CF is an important risk factor for severe infection by influenza [23], and adequate treatment and prevention of this condition must be encouraged.

In contrast to other studies which showed PE to be more severe in cases with viral detection [3, 8, 14, 24], our findings didn't show a significant difference between virus positive and negative groups, measured by the CFCS and by dyspnea, low oxygen and hospital admission rates. However, median age in these studies was older, including adolescents and adults, and it must also be stressed that the limited size of the sample may have impacted the power of statistical analysis.

Viral infections have been shown to increase susceptibility to bacterial colonization of the respiratory tract [25, 26]. Since this colonization plays a major role in the progression of lung disease in CF, the relationship between these agents in CF is an important object of study. RSV has been shown to promote *Pseudomonas* colonization in CF patients [27, 28], and the same has been suggested for *picornavirus* [11]. In patients already chronically colonized by this bacteria, RV may increase liberation of planktonic bacteria from the biofilm [29] which is associated with new bacterial infection. On the other hand, a study by Chin *et al* didn't find an increase in *Pseudomonas* density in sputum during viral infections [30]. As in most previous studies [9, 10, 22, 31] we found no difference of pseudomonas prevalence in patients with viral detection.

It has also been suggested that the presence of bacteria in the respiratory tract of CF patients may favor viral infection [25], but more studies regarding the specific role of *Pseudomonas* in this interaction are still needed. In the present study, virus detection did not significantly differ between different types of previous bacterial colonization. It is important to consider that in developing countries such as Brazil, the age of *Pseudomonas* colonization tends to be younger than in wealthier countries where most of these cited studies were performed. In our population, 57,1% of samples belonged to children with previous intermittent or chronic *P. aeruginosa* colonization.

The prevalence of respiratory virus, although significant, was smaller than previously reported in some prior studies that also applied molecular technics [3–6, 11]. An explanation for this difference is the broader PE definition criteria [3, 5, 6] and the inclusion of samples from patients with upper respiratory tract infections without PE [4, 11]. In our perspective, a

more open inclusion criteria eventually included milder cases, less important for the course of the disease or the clinical conduct.

In order to account for viral seasonality patterns, samples were collected through 12 consecutive months. On the other hand, some studies were conducted only throughout autumn and winter seasons, when there's a higher circulation of respiratory virus, which may contributed to higher viral detection rates [3, 4, 7].

The study's main limitations were the size of the sample and timing of sample collection. In a previous study in which patients were trained to self-collect samples as soon as the first signs of respiratory disease appeared and mail them to the laboratory, virus detection rates reached 81% of 43 samples investigated [6].

In addition, we did not test for coronavirus or bocavirus. Even though these agents have been usually shown to have a small prevalence in PE [3, 5–7, 10, 12, 20], this may have had some impact on the total viral detection rates, and especially on the seven samples in which neither virus nor bacteria were identified.

## Conclusions

Our findings corroborate that respiratory virus have a significant prevalence in pulmonary exacerbations in CF. Their detection was associated with change in sinus discharge and in children <5 years with fever. Routine testing for these agents may help to better guide PE treatment and antibiotic use. Furthermore, as PEs have a great impact on morbidity and mortality in CF patients, the recognition of respiratory virus as an important agent in these conditions proves how fundamental it is to increase preventive strategies such as isolation protocols for patients and immunization. Unfortunately, there still are no effective vaccines, prophylaxis and treatments for most respiratory virus, however the viral diagnostic supported the rational use of antibiotics, avoiding its misuse. Longitudinal studies are still needed to better understand the relationship between virus and bacterial colonization in CF.

## Supporting information

**S1 Appendix. De-identified data set.**
(XLSX)

## Acknowledgments

For their support, we thank all the researchers and technicians in Respiratory Virus and Measles Laboratory at IOC/FIOCRUZ, and all the physicians in the cystic fibrosis center of Fernandes Figueira Institute.

## Author Contributions

**Conceptualization:** Viviane Mauro Correa Meyer, Marilda Mendonça Siqueira, Patricia Fernandes Barreto Machado Costa, Tânia Wrobel Folescu, Fernando do Couto Motta.

**Data curation:** Viviane Mauro Correa Meyer, Patricia Fernandes Barreto Machado Costa, Jonathan Christian Oliveira Lopes, Fernando do Couto Motta.

**Formal analysis:** Viviane Mauro Correa Meyer, Braulia Costa Caetano, Jonathan Christian Oliveira Lopes, Fernando do Couto Motta.

**Funding acquisition:** Marilda Mendonça Siqueira, Fernando do Couto Motta.

**Investigation:** Viviane Mauro Correa Meyer, Patricia Fernandes Barreto Machado Costa, Braulia Costa Caetano, Jonathan Christian Oliveira Lopes, Tânia Wrobel Folescu.

**Methodology:** Viviane Mauro Correa Meyer, Marilda Mendonça Siqueira, Patricia Fernandes Barreto Machado Costa, Tânia Wrobel Folescu, Fernando do Couto Motta.

**Project administration:** Viviane Mauro Correa Meyer, Marilda Mendonça Siqueira, Patricia Fernandes Barreto Machado Costa, Jonathan Christian Oliveira Lopes, Fernando do Couto Motta.

**Resources:** Braulia Costa Caetano.

**Supervision:** Patricia Fernandes Barreto Machado Costa, Fernando do Couto Motta.

**Validation:** Jonathan Christian Oliveira Lopes.

**Writing – original draft:** Viviane Mauro Correa Meyer.

**Writing – review & editing:** Viviane Mauro Correa Meyer, Marilda Mendonça Siqueira, Patricia Fernandes Barreto Machado Costa, Braulia Costa Caetano, Jonathan Christian Oliveira Lopes, Tânia Wrobel Folescu, Fernando do Couto Motta.

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
