## [Decision Letter · Decision Letter 0]

17 Aug 2020

PONE-D-20-16036

Clinical impact of respiratory virus in pulmonary exacerbations of children with Cystic Fibrosis

PLOS ONE

Dear Dr. Viviane Mauro Correa Meyer,

Thank you for submitting your manuscript to PLOS ONE. After careful consideration, we feel that it has merit but does not fully meet PLOS ONE’s publication criteria as it currently stands. Therefore, we invite you to submit a revised version of the manuscript that addresses the points raised during the review process.

We look forward to receiving your revised manuscript.

Kind regards,

Abdelwahab Omri, Pharm B, Ph.D

Academic Editor

PLOS ONE

Additional Editor Comments:

This is a very simple retrospective analysis of data that I think is flawed in combining static, patient-specific features with variable event-specific features. I would encourage evaluation by a statistician.

Journal Requirements:

Reviewers' comments:

Reviewer's Responses to Questions

**Comments to the Author**

1. Is the manuscript technically sound, and do the data support the conclusions?

Reviewer #1: Partly

Reviewer #2: Partly

Reviewer #3: Partly

Reviewer #4: Partly

2. Has the statistical analysis been performed appropriately and rigorously? 

Reviewer #1: Yes

Reviewer #2: Yes

Reviewer #3: I Don't Know

Reviewer #4: Yes

3. Have the authors made all data underlying the findings in their manuscript fully available?

Reviewer #1: No

Reviewer #2: No

Reviewer #3: Yes

Reviewer #4: Yes

4. Is the manuscript presented in an intelligible fashion and written in standard English?

Reviewer #1: Yes

Reviewer #2: No

Reviewer #3: Yes

Reviewer #4: Yes

5. Review Comments to the Author

Reviewer #1: Although the manuscript is an original work, it is more like an observational study, in which the findings reported are very similar to previously published articles, at least in some form.

The conclusion of this study is very similar to other published reports and thus it does not fulfill the criteria of Replication studies described in the Plos One Guidelines for Reviewers.

I suggest to submit it as an 'Observational study' instead of 'original article'

Reviewer #2: The authors have completed a prospective collection of samples and data on a cohort of subjects with cystic fibrosis. They have determined the viral infection at the time of pulmonary exacerbation and identified factors that differentiate those that are virus positive compared with virus negative.

The authors have provided a comprehensive introduction and discussion with reference to the established literature. They authors have used well established assessment tools in CF research. As such, the study provides some useful information.

I have a few major and minor comments to make that could be addressed by the authors.

Major

1. Since the number of positive subjects is quite limited, I would like to know the power to detect statistical differences between virus positive and negative groups for each outcome. This could be added as an online supplement. I am concerned this study does not have the statistical power to detect differences based on the subject numbers and that is why so few outcomes are significant.

2. If a result is not statistically significant (or even close to p-0.05) then there is no relationship. For example, the authors claim that they virus positive group were younger, but statistically they were not (p=0.3), therefore they are NOT younger. This is described in the results and the discussion. This also includes fever, fatigue antibiotic prescription, P. aeruginosa prevalence, acquisition of P. aeuginosa, intermittent or chronic P. aeruginosa in the whole group or the children <5 years of age (outlined in the discussion).

3. It would be helpful if the authors identified which previous studies mentioned in the discussion (for example: when the authors are discussing the lack of difference in CF severity between virus positive and negative groups) were of a similar age and which were from older children.

Minor

1. Although the article is quite well written and perfectly understandable – the article requires further editing for English.

2. It would be helpful if standard acronyms were used such as URTI (upper respiratory tract illness) instead of URTS.

Reviewer #3: The manuscript by Correa Meyer et al describes an analysis of respiratory virus infection in persons with CF during episodes of pulmonary exacerbation (PE). The aim of the study was to compare virus-positive and virus-negative groups in regards to clinical manifestations, severity of PE and bacterial infection. The authors conclude that prospective studies are needed to assess the impact of viral infection in CF.

A concern with the design and analysis of data in this study is that patient-specific (static) and exacerbation event-specific (variable) characteristics are combined and compared (Table 2). That is, some patient-specific features (eg CFTR genotype, gender pancreatic insufficiency) are counted as independent features for each episode of exacerbation. This seems to introduce confounding. Eg, if an individual heterozygous for F508del had three exacerbation episodes, but by virtue of lack of siblings or age, none of these were associated with viral infection, the three episodes would inflate the number of virus-negative events in the F508del heterozygous group… supporting that viral infection is less common in persons heterozygous for F508del. A more robust multivariable analysis with attention to interactions between variables seems to be required.

On a more minor note: There are implications of causality, when associations are all that can be assessed. Eg: Abstract: do exacerbations cause morbidity in CF … or are these events a sign of morbidity? Eg: Intro: do exacerbations impact disease progression…or are these events a manifestation of lung disease progression? Eg, have the studies cited (refs 3-15) show that viral infection is ‘responsible for’ exacerbation events?

Reviewer #4: This is a study from Brazil in pediatric patients with CF that explores whether viral DNA/RNA can be found during a CF exacerbation. The authors found that 36% of 70 respiratory CF exacerbations were associated with retrieval of viral genetic material from NP swabs.

Comments:

1) Unfortunately this is a relatively small study, and statistical power is quite limited. Therefore the investigators were not able to show significant differences between viral-associated exacerbations and non-viral exacerbations.

2) I think that the study abstract and conclusions need to stress lack of statistical power as a major limitation of this study, this may explain why significant differences between viral-associated exacerbations and non-viral exacerbations was not found.

3) The fact that viral exacerbations were associated with upper resp tract symptoms is of course not surprising, and this may need to be made clearer in the abstract and conclusions.

4) Tables 2 and 5 are far too long and hard to read and all statistical comparisons are non-significant. Please limit these tables to important variables only- please shorten them to 5-6 variables.

5) The discussion should include mention of a study that is highly relevant : Chin M, De Zoysa M, Slinger R, Aaron SD. Acute effects of viral respiratory tract infections on sputum bacterial density during CF pulmonary exacerbations. Journal of Cystic Fibrosis, 14: 482-489, 2015.

6. PLOS authors have the option to publish the peer review history of their article (what does this mean?). If published, this will include your full peer review and any attached files.

Reviewer #1: No

Reviewer #2: No

Reviewer #3: No

Reviewer #4: No

---

## [Author Response · Author response to Decision Letter 0]

21 Sep 2020

We thank Reviewer #1 for their considerations, but we checked with the journal office and PLOS ONE does not have a 'Observational study' article type. Therefore we resubmitted the manuscript as a "Research Article", which include observational studies. 

We appreciate Reviewer #2 for their considerations. The study was reviewed by a statistician and we have included a minimal anonymized data set from the Study as a Supporting Information File. The discussion was also rewritten. All sections describing relationships that were not statistically significant were reviewed and corrected to show that in fact no relationship could be found. The limited size of the sample was also stressed as a potential limitation for statistical analysis. The manuscript was also thoroughly reviewed and edited for English. 

We appreciate reviewer #3's comments. In fact, we reviewed the study with a statistician and excluded patient-specific features from the comparison of virus positive and virus negative exacerbations. These features were separately described in Table 2. Due to the limited size of the sample a more robust multivariable analyses was not possible.

We thank Reviewer #4 for their comments. We included the suggested article in the discussion, shortened the tables and stressed potential limitations of the study.

---

## [Editor Report · Decision Letter 1]

28 Sep 2020

Clinical impact of respiratory virus in pulmonary exacerbations of children with Cystic Fibrosis

PONE-D-20-16036R1

Dear Dr. Viviane Mauro Correa Meyer,

We’re pleased to inform you that your manuscript has been judged scientifically suitable for publication and will be formally accepted for publication once it meets all outstanding technical requirements.

Kind regards,

Abdelwahab Omri, Pharm B, Ph.D

Academic Editor

PLOS ONE

---

## [Editor Report · Acceptance letter]

7 Oct 2020

PONE-D-20-16036R1 

Clinical impact of respiratory virus in pulmonary exacerbations of children with Cystic Fibrosis 

Dear Dr. Meyer:

I'm pleased to inform you that your manuscript has been deemed suitable for publication in PLOS ONE. Congratulations! Your manuscript is now with our production department. 

Kind regards, 

on behalf of

Dr. Abdelwahab Omri 

Academic Editor

PLOS ONE